# Tailoring baker's yeast *Saccharomyces cerevisiae* for functional testing of channelrhodopsin

Sebastian Höler[1], Daniel Degreif[1], Florentine Stix[1], Shang Yang[2], Shiqiang Gao[2], Georg Nagel[2], Anna Moroni[3], Gerhard Thiel[1,4], Adam Bertl[1,4], Oliver Rauh[1,4]*

1 Department of Biology, Technische Universität Darmstadt, Darmstadt, Germany, 2 Institute of Physiology–Neurophysiology, Biocentre, Julius-Maximilians-University, Wuerzburg, Germany, 3 Department of Biosciences and CNR IBF-Mi, Università degli Studi di Milano, Milano, Italy, 4 Centre for Synthetic Biology, Technische Universität Darmstadt, Darmstadt, Germany

* rauh@bio.tu-darmstadt.de

**Data Availability Statement:** All relevant data are within the paper and its Supporting information files.

**Funding:** GT; AM: European Research Council GT: German Research Foundation. GN: German Research Foundation. The funders had no role in

## Abstract

Channelrhodopsin 2 (ChR2) and its variants are the most frequent tools for remote manipulation of electrical properties in cells via light. Ongoing attempts try to enlarge their functional spectrum with respect to ion selectivity, light sensitivity and protein trafficking by mutations, protein engineering and environmental mining of ChR2 variants. A shortcoming in the required functional testing of large numbers of ChR2 variants is the lack of an easy screening system. Baker's yeast, which was successfully employed for testing ion channels from eukaryotes has not yet been used for screening of ChR2s, because they neither produce the retinal chromophore nor its precursor carotenoids. We found that addition of retinal to the external medium was not sufficient for detecting robust ChR activity in yeast in simple growth assays. This obstacle was overcome by metabolic engineering of a yeast strain, which constitutively produces retinal. In proof of concept experiments we functionally express different ChR variants in these cells and monitor their blue light induced activity in simple growth assays. We find that light activation of ChR augments an influx of $Na^+$ with a consequent inhibition of cell growth. In a $K^+$ uptake deficient yeast strain, growth can be rescued in selective medium by the blue light induced $K^+$ conductance of ChR. This yeast strain can now be used as chassis for screening of new functional ChR variants and mutant libraries in simple yeast growth assays under defined selective conditions.

## Introduction

In one approach of optogenetics natural or synthetic light gated ion channels are expressed in cells of interest, where they can be activated or inhibited by wavelength-specific light stimuli [1, 2]. This offers the possibility of non-invasive modulation of activity in the targeted channels with high temporal and spatial precision. By using this remote control over channel activity, the electrical properties of cells can be precisely tuned in experimental research and even for biomedical applications [3–8].

study design, data collection and analysis, decision
to publish, or preparation of the manuscript.

**Competing interests:** The authors have declared
that no competing interests exist.

The first and so far most frequently used tools in this type of optogenetics are channelrhodopsins [9–11]. The originally discovered channelrhodopsins 1 and 2 (ChR1 and ChR2) from the green alga *Chlamydomonas rheinhardtii* and their currently known variants have structural similarities to microbial ion pump rhodopsins, but unlike the latter, ChRs act directly as light-activatable cation channels [9, 10]. The original ChR2 absorbs blue light with a maximum at 460 nm *via* an all-*trans* retinal chromophore in complex with the channel protein. Upon light absorption the chromophore undergoes a *trans* to *cis* isomerization, which eventually leads to channel opening. The open channel is cation-selective and transports monovalent ($H^+$, $Na^+$, $K^+$, etc.) and divalent (e.g. $Ca^{2+}$) ions [10, 12].

Increasing sophistication of optogenetic applications poses a growing demand for light gated channels with different functional properties. This includes different types of ion selectivity, shifts of action spectra [13, 14] as well as an elevated sensitivity to light [15]. Also channels with different kinetic properties for long timescale experiments or high precision optogenetic control [16] or proteins with improved sorting to the target membrane are required [17]. Intensive work on ChRs over the last two decades has shown that many of the desired features like anion selectivity or increased $K^+$ selectivity can already be found in some exotic natural occurring variants of this protein [18, 19]. Alternatively, they can be achieved by rational design and mutagenesis of existing ChRs [20, 21]. Altogether these findings suggest that ChRs have a high degree of plasticity with respect to structure/function correlates. Hence it is reasonable to speculate that still more functional features can be achieved via protein engineering [22] or environmental mining [22, 23]. The electrophysiological characterization of ChRs variants and mutants in oocytes or mammalian cells is this context still the gold standard. But while this works well for small numbers of candidate proteins a screening of large numbers of mutants or variants would benefit from a simple high throughput screening system. This would provide a fast way of identifying candidate proteins with new and interesting functional features worth pursuing in a more detailed analysis. For many ion channels from eukaryotic cells such a simple screening system is offered by baker's yeast *Saccharomyces cerevisiae* [24]. This unicellular eukaryotic organism proved in the last two decades an optimal platform for a rapid testing of channel activity in newly discovered channel proteins [25], functional screening of channel mutants [26] and testing of ion selectivity as well as drug sensitivity [27, 28]. The same system was also useful for a rapid screening of efficient sorting of channel proteins to the plasma membrane [29] as well as for functional testing of *de novo* engineered channel proteins [30]. ChRs have so far already been expressed in yeasts like *Komagatella phaffii* (aka *Pichia pastoris*) and *S. cerevisiae* for producing large amounts of proteins [31, 32]. But to our knowledge, there is only one publication reporting expression and functional studies of ChR2 in *S. cerevisae*. ChR2-mediated currents were in this study monitored in patch-clamp recordings from giant yeast protoplasts produced by electrofusion [33]. Apart from this study, which seems in terms of the experimental approach much too complex for employing it in functional screening experiments, there are no reports yet on a functional testing of ChRs in yeast in general or in *S. cerevisiae* in particular. One obstacle for using ChRs in *S. cerevisiae* is that the yeast is neither producing the chromophore all-*trans* retinal nor carotenoids as a precursor. This limitation can be overcome in some cases like in *Xenopus* oocytes by adding the membrane permeant retinal to the extracellular solution [34] or in plants by introducing retinal-producing genes [35]. Studies on functional reconstitution of opsins in *S. cerevisiae* however have shown that only a small fraction of the total expressed recombinant opsin exhibited an integration of the chromophore into the protein [36]. Hence it is not clear if an extracellular supply of the chromophore is sufficient for generating functional ChRs in *S. cerevisiae*.

Here, we show that application of retinal into the external medium was not sufficient for guaranteeing ChR activity sufficient to trigger growth phenotypes in baker's yeast. By different

steps of metabolic engineering, we were however able to generate a yeast strain, which constitutively produces the retinal chromophore, which is essential for ChR activity. This yeast strain can be used as a chassis to successfully express ChRs and test their light stimulated function in simple yeast growth assays.

## Results and discussion

In order to generate a detectable phenotype in yeast growth the light-activated Channelrhodopsin 2 (ChR2) must be localized in the plasma membrane. To test synthesis and sorting of ChR2 in yeast, an eYFP-tagged version of the protein was expressed under control of an inducible *GAL1* promotor. Confocal images from an overnight culture show that ChR2 is synthesized in yeast in appreciable amounts. The fluorescent signal is visible in endo- membranes, presumably the endoplasmic reticulum, as well as in the plasma membrane (Fig 1A). Since yeast cells require only a low concentration of channels in their plasma membrane for ion uptake, a detectable GFP signal from ChR2 in the plasma membrane should hence allow functional testing of the light gated channel in yeast growth assays. Indeed, control experiments (S1 Fig) confirm that a low fluorescent signal form a channel protein in the plasma membrane of yeast cells is sufficient for employing them in complementation assays. The sensitivity of these complementation assays to membrane impermeable blockers (e.g. [27]) underpins that the complementation assays are determined by the plasma membrane resident protein and not by protein in endo-membranes.

A simple test could be based on a light induced inhibition of yeast growth by $Na^+$. Even though elevated concentrations of $Na^+$ in the cytosol are toxic, wild type yeast cells generally tolerate high sodium concentrations of $>1M$ in the external medium [37]. This tolerance should be decreased when cells express a protein, which elevates the otherwise low $Na^+$ conductance of the plasma membrane. To test this prediction, we expressed the $Na^+$ permeable capsaicin receptor TRPV1 in yeast. The growth assay in Fig 1B confirms that activation of this $Na^+$ conducting channel by capsaicin causes a strong inhibition of yeast growth in a medium with a high $Na^+$ concentration. Function of TRPV1 and plasma membrane localization in this yeast strain was also confirmed by capsaicin triggered calcium increase monitored in a aequorin based luminometric assay (Fig 1C). This capsaicin induced calcium signal was absent in the yeast strain not expressing TRPV1. We hypothesized that activation of the $Na^+$ conducting ChR2 by light should render the growth of yeast expressing this protein in the same manner sensitive to high extracellular $Na^+$.

To test the function of the light gated channel in such a simple growth assay, yeast cells transformed with ChR2 or its D156C mutant, which shows increased open probability in the light [38], were grown on selective agar plates (SGal-ura) ± 500 mM NaCl (Fig 1D). In the absence of 500 mM $Na^+$, BY4741 cells transformed with the empty plasmid (ev) showed equally robust growth in the dark and in the light, and no difference was observed for cells expressing ChR2 or its D156C mutant. The presence of 500 mM NaCl in the agar caused a slight growth inhibition in all three yeast strains, which is most evident by comparing colonies at the 1000-fold dilution (Fig 1D); but as in the absence of $Na^+$, there was no light effect detectable. Since yeast cells neither produce carotenoids as precursor nor retinal as essential chromophore for ChR2 function, agar plates containing 500 mM $Na^+$ were additionally supplemented with 10 μM retinal, to allow the yeast cells to take up the chromophore and incorporate it into ChR2. In contrast to our expectation, yeast grown for three days in SGal-ura + 500 mM $Na^+$ showed no light sensitivity, regardless of the presence or the absence of retinal. The data in Fig 1D hence underscore a general weak sensitivity of the cells to 500 mM NaCl stress, but this stress is not augmented by conditions which should promote ChR2 or ChR2(D156C) activity

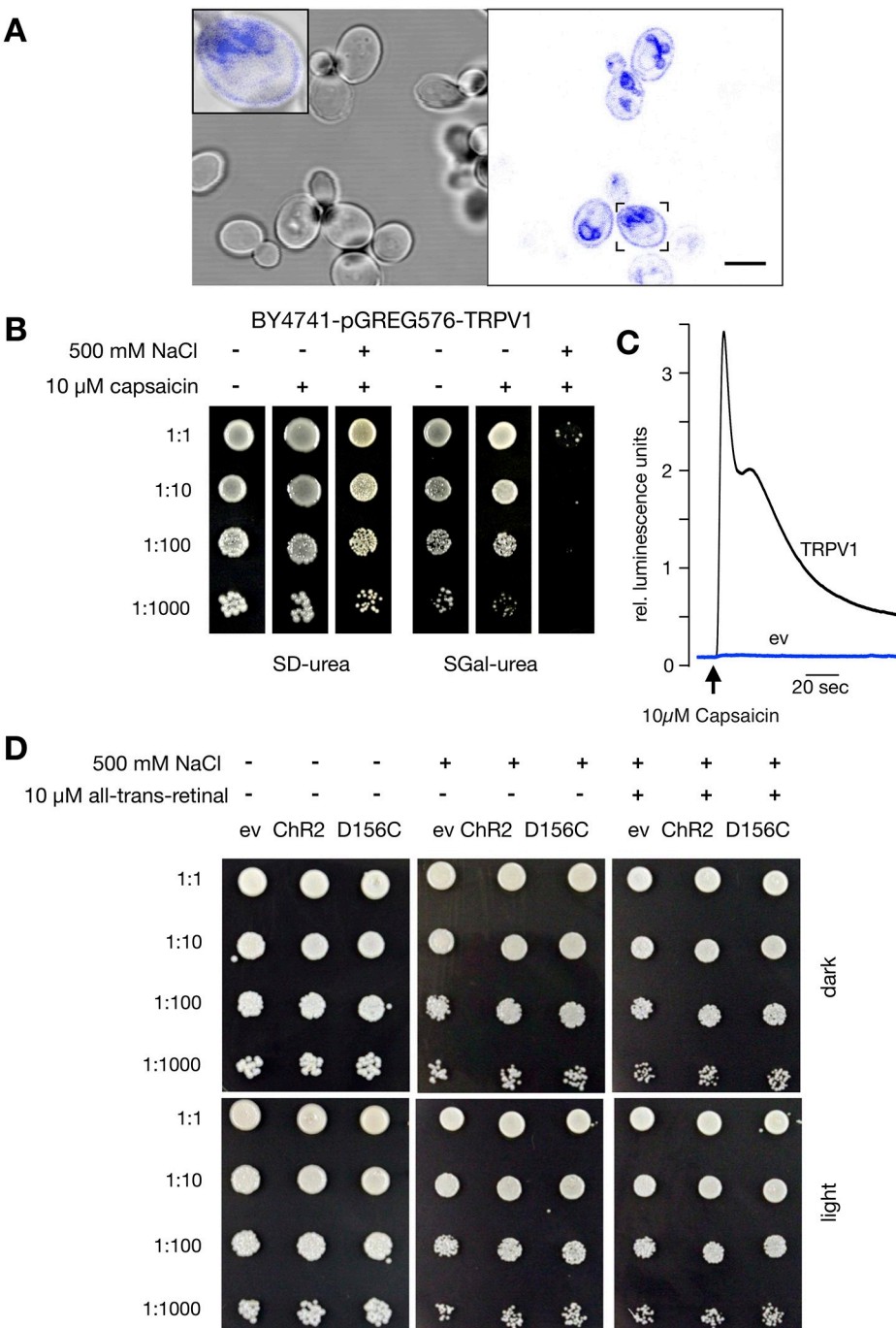

**Fig 1. ChR2 is properly expressed in yeast strain BY4741 but shows no sign of function. (A)** Confocal images of BY4741 yeast strain in SGal-ura overnight cultures. Transmission images (left) and corresponding fluorescent images (right) of eYFP-tagged ChR2. *Inset*: Magnification and overlay of transmission and fluorescent images from cell indicated in right panel showing that fluorescence is visible in position of the plasma membrane. **(B)** Droplet test of BY4741 transformed with the pGREG576 plasmid bearing the ORF for the capsaicin receptor TRPV1 from *Rattus norvegicus*. In SD-ura, without induction of transcription, the presence of either the channel activator capsaicin, the permeant $Na^+$ or both had very little effect on growth (left panel). With induction of transcription by galactose (right panel), channel activation by capsaicin alone had little effect on growth. However, the additional presence of 500 mM NaCl resulted in a strong growth inhibition. **(C)** Luminometric assay reveals capsaicin triggered calcium influx in BY4741 yeast cells expressing TRPV1, but not in control cells harbouring the empty pGREG576 plasmid. Arrow indicates injection of 10 μM capsaicin. Curves are the means of 6 recordings for the TRPV1 strain and 2 recordings for the empty vector control. **(D).** Droplet test of BY4741 transformants without (-) and with (+) 500 mM NaCl salt stress.

Cells grown in absence (-) and presence (+) of 10 µM retinal for three days in dark and light. Cells harboring the empty pGREG506 plasmid (ev) and cells expressing either ChR2 (ChR2) or its D156C mutant (D156C) where spotted as serial dilutions from an $OD_{600}$ = 1 on SGal-ura agar plates at 10 µL per spot.

in the plasma membrane. Therefore, we concluded that the conductance of ChR2 and its D156C mutant was too small or not sufficiently activated under the experimental conditions tested.

One possible explanation for this negative result is that retinal might not be sufficiently taken up for rendering ChR2 light sensitive. During incubation of cells in the light with externally supplied retinal at a rather high concentration, we observed a decrease in the intensity of the yellow color of the agar medium. To test if this can be interpreted as evidence for an uptake of retinal into the cells, experiments from Fig 1B were repeated without cells. The images in Fig 2A show agar plates with 1 mM retinal before and after incubation for 24 h in either darkness or light. The representative images and the corresponding analysis of the yellow color values (Fig 2A and 2B) exhibit a visible bleaching only after light exposure. Hence a decrease in color intensity is not evidence for retinal uptake into cells but rather an indication for a light-dependent photo-destruction of the chromophore.

## Engineering of an all-trans retinal producing *trk1Δtrk2Δ* yeast strain

For engineering a versatile yeast chassis strain for screening ChR function, we decided to use the PLY240 strain [37], which is devoid of the major $K^+$ uptake systems Trk1p and Trk2p and thus suitable for studying complementation of $K^+$ uptake deficiency by heterologous $K^+$ transporters. In addition, the absence of Trk1p and Trk2p offers the benefit of an increased sensitivity towards toxic external cations such as $Na^+$, $Li^+$ or $Cs^+$ [37, 39]. To overcome the absence of retinal in yeast, we modified the PLY240 strain for constitutive synthesis of this essential cofactor for ChR. In an initial step three genes (crtI, crtYB and crtE) from the carotenoid-producing yeast *Xanthophyllomyces dendrorhous*, which enable β-carotene synthesis in *S. cerevisiae* cells [40], were genomically integrated in a marker-free manner. For this, a polycistronic construct [41] was integrated into the *CAN1* locus of the PLY240 (*trk1Δtrk2Δ*) yeast strain [37] by means of the CRISPR/Cas9 technology [42] resulting in strain SHY1. Integration of these genes generated yellow-colored yeast colonies, indicating successful production of β-carotene in the engineered cells (Fig 3A, left agar plate). In the next step a carotene monoxygenase (carX) from the ascomycete *Fusarium fujikuroi* [43] was genomically integrated into the *kanMX* cassette (remaining from the trk2-deletion in PLY240) of the SHY1 strain to give strain SHY2. This resulted in a weaker yellow color of the colonies (Fig 3A, right agar plate) suggesting cleavage of β-carotene to retinal by the carX activity.

However, UPLC analysis of organic phase extracts from these cells did not detect all-*trans* retinal suggesting that the β-carotene synthesis is insufficient to form enough precursors for robust all-*trans* retinal synthesis. Therefore, a second crtI gene was integrated into the *YPRCΔ15* locus [44] of the aforementioned β-carotene producing cells (SHY1) to give strain SHY3. We reasoned that this might increase biosynthesis of β-carotene [40]. Indeed, after genomic integration of the polycistronic β-carotene synthesis pathway plus a second crtI gene the SHY3 colonies showed a more intense orange color (Fig 3B, left agar plate) indicating enhanced carotenoid production.

Following genomic integration of carX into the *kanMX* cassette of the SHY3 cells, the colonies of the resulting SHY4 strain lost their intense orange color implying efficient β-carotene cleavage and synthesis of retinal. (Fig 3B, right agar plate). This was confirmed by UPLC

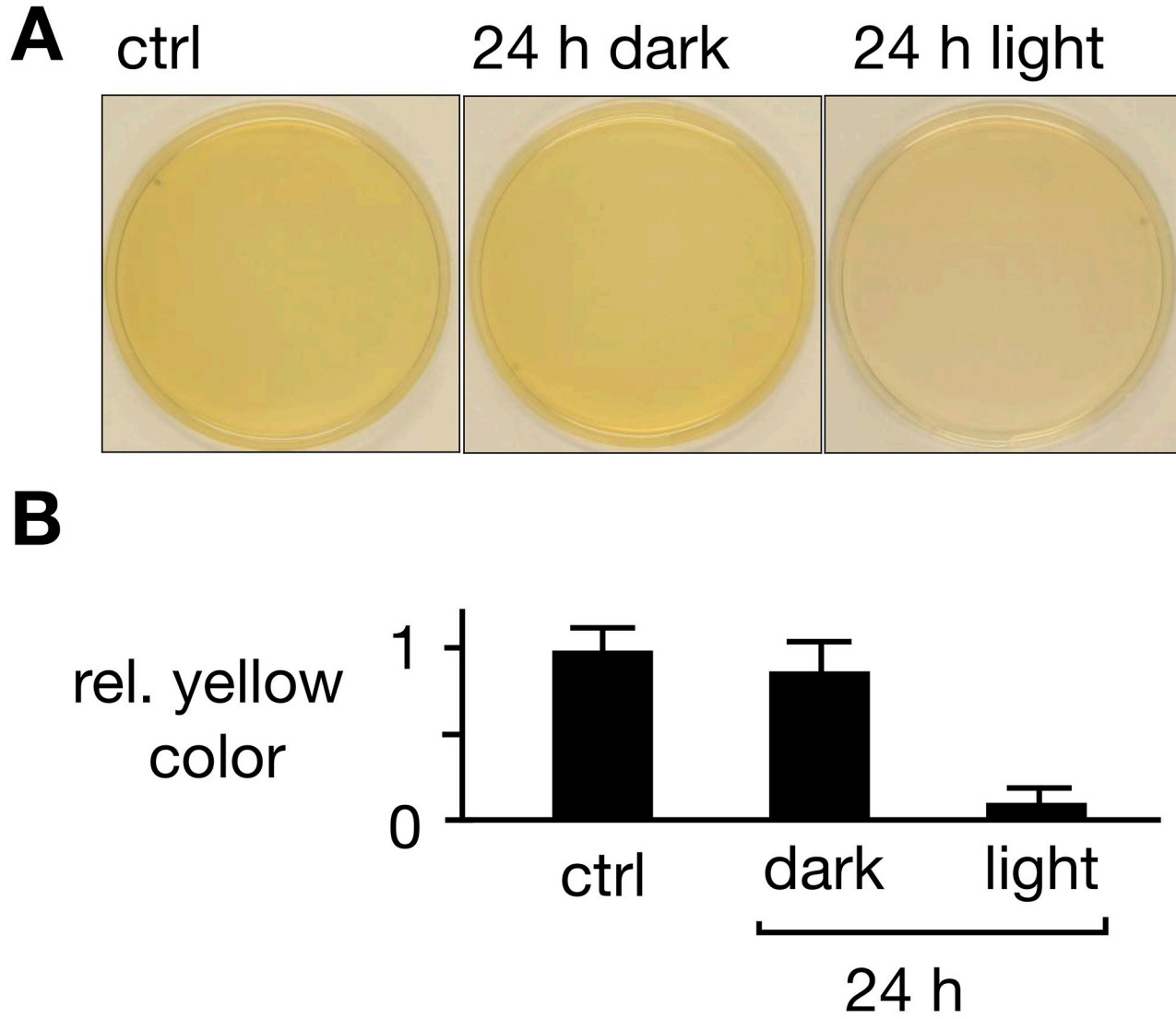

**Fig 2. Illumination bleaches retinal in medium. (A)** Agar plates containing 1 mM retinal directly after preparation (left, ctrl) and 24 h after keeping plate in the dark (center, 24 h dark) or in the light (right, 24h light, right). Illumination bleaches retinal-containing agar. **(B)** Yellow color of agar plate above background was obtained from deconvolution of images and plotted as relative value of plates after 24 h incubation in light or dark relative to pre-incubation (ctrl). Mean ± SD of independent replicates.

analysis of an ethanol cell extract, which shows the same elution peak as a 5 µM all-*trans* retinal reference sample (Fig 3C).

We reasoned that a ChR variant with higher conductance for Na⁺ and K⁺ might generate a stronger phenotype than wild type ChR2. Thus, a ChR variant, named ChR2-5x, with improved Na⁺ and K⁺ conductance (Fig 4A, [45]) was selected to perform further studies.

When ChR2-5x was expressed in *Xenopus* oocytes and activated by blue light an elevation of extracellular Na⁺ or K⁺ from 1 mM to 120 mM generated a positive shift of the reversal potential by 91.5 mV in NaCl and 81 mV in KCl containing solution (Fig 4B). The amplitude of the shift is higher in the case of ChR2-5x than with the ChR2-XXM variant [46] (Fig 4B). The same is true for experiments with wt-ChR2, which exhibits in the same assay only a moderate positive shift of 24 mV in Na⁺ and 21 mV in K⁺ [47]. The results of these experiments

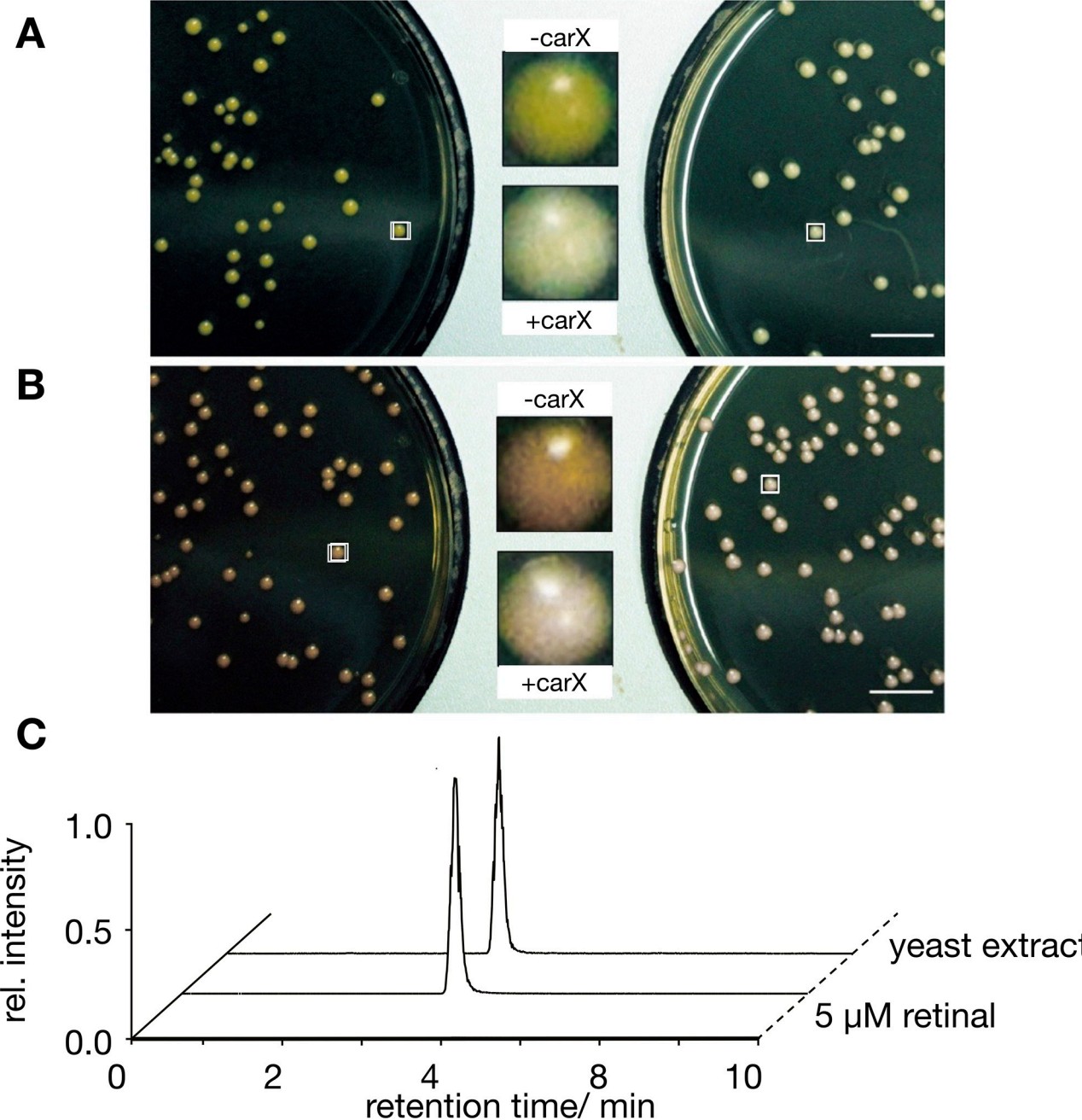

**Fig 3. Engineering of retinal producing yeast. (A)** PLY240 cells with genomic integration of the polycistronic construct for β-carotene production without (left) and with the carotenoid monooxygenase carX (right) were plated at $1:10^4$ dilution on nonselective medium containing a high $K^+$ concentration to yield single colonies. **(B)** *trk1Δtrk2Δ* cells with genomic integration of the polycistronic construct for β-carotene production and a second crtI gene without (left) and with (right) the carotenoid monooxygenase carX plated as in **A**. The colonies highlighted by frames are magnified and shown in the center. Images in **A** and **B** were taken after 4 days of incubation. Scale bar = 10 mm. **(C)** UPLC analysis of retinal production in yeast. Ethanol extract from *trk1Δtrk2Δ* yeast cells as in **B** with the genomically integrated polycistronic β-carotene pathway, a second crtI gene and carX was used for UPLC analysis with 5 μM all-*trans* retinal as reference. Data were normalized to the largest peak in the chromatograms. The analysis shows the same elution behavior of the extract as for the reference with a signal at 3.54 min indicating production of all-*trans* retinal in yeast cells.

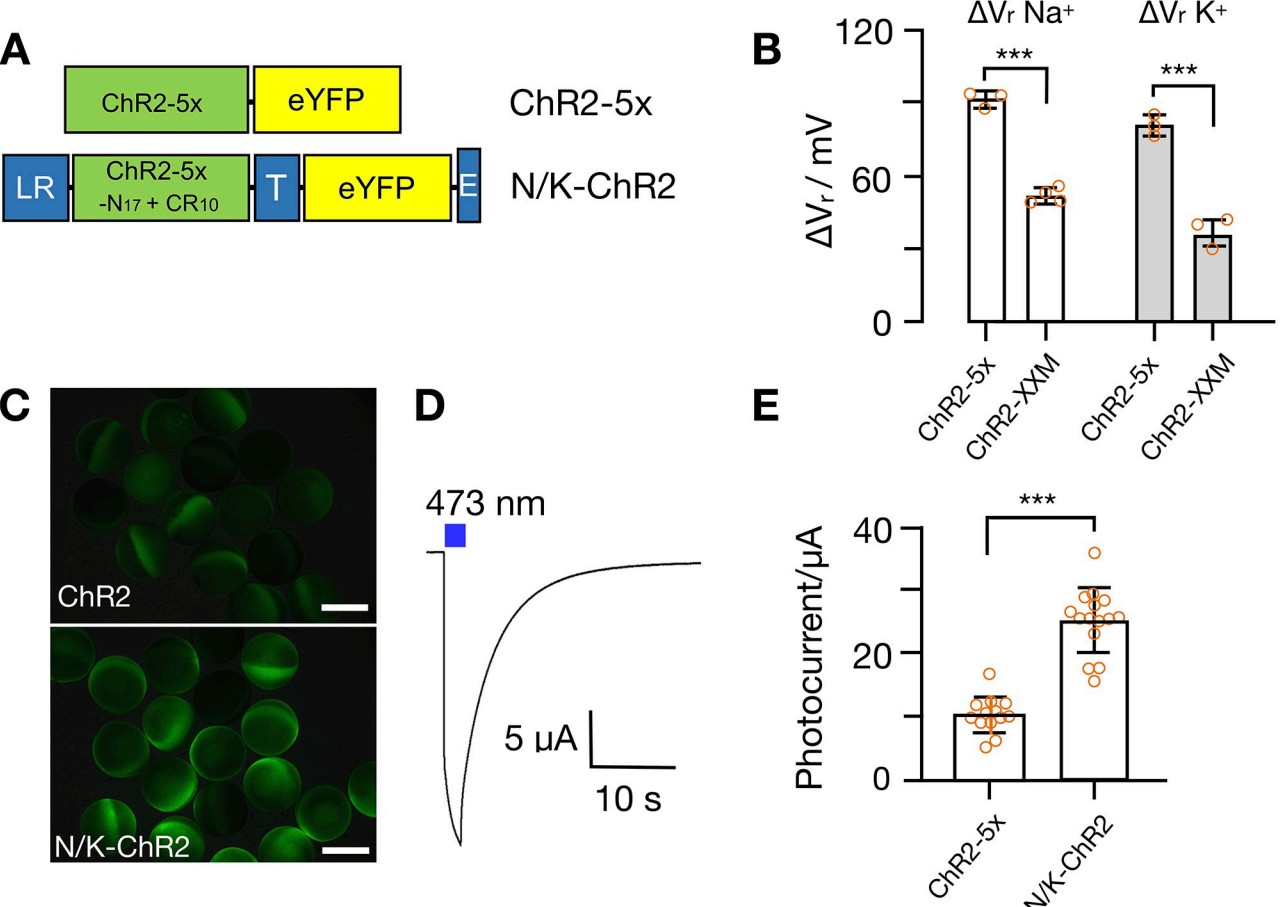

**Fig 4. Engineering of N/K-ChR2. (A)** Schemes of ChR2-5x and N/K-ChR2. ChR2-5x is a ChR2 variant with 5 point mutations. N/K-ChR2 comprises ChR2-5x scaffold with $CR_{10}$ (= first 10 amino acids from proton pump rhodopsin CsR) and $N_{17}$ (= first 17 amino acids from ChR2). The construct contains additional domains in N- and C-terminus namely LR domain (= N-terminal signal peptide Lucy-Rho), T domain (= the plasma membrane trafficking signal from Kir2.1) and E domain (= the endoplasmic reticulum (ER) export signal from Kir2.1). **(B)** Shift in reversal potential ($\Delta V_r$) recorded in *Xenopus* oocytes expressing either ChR2-5x or ChR2-XXM upon changing extracellular solution from 1 mM to 120 mM NaCl ($\Delta V_r$ Na$^+$) or KCl ($\Delta V_r$ K$^+$). Values represent mean ± SD, n = 3–4. **(C)** Images of *Xenopus* oocytes expressing ChR2-5X and N/K-ChR2; scale bar = 1 mm. **(D)** Representative photocurrent trace from N/K-ChR2 expressing oocyte elicited by illumination with blue light (472 nm, blue bar); oocyte incubated in solution containing (in mM) 110 NaCl, 5 KCl, 2 BaCl$_2$, 1 MgCl$_2$, 5 HEPES/ pH 7.6). Holding potential at -70 mV. **(E)** Mean photocurrent amplitudes of ChR2-5x and N/K-ChR2 recorded as in **C**; n = 13–15. P value of statistical significance in **B** and **E** from unpaired t-test indicated as *** = P < 0.001.

indicate an enhanced selectivity of the channel for K$^+$ and Na$^+$ over other competing cations. Additional modifications of the N- and C-termini of ChR2-5x, resulting in the variant N/K-ChR2 (Fig 4A), further increased the expression level and/or membrane trafficking of ChR2-5x (Fig 4C). This is apparent in a higher GFP fluorescence in oocytes (Fig 4C) and reflected in a significantly higher photocurrent amplitude of N/K-ChR2 compared to ChR2-5x (Fig 4E). When the optimized ChR2-5x, termed N/K-ChR2, was expressed in SHY4 cells, we found that the fluorescent signal of the GFP tag was, like in Fig 1, visible in some endo-membranes but also as ring at the periphery of cells (Fig 5A); in many cells the fluorescence was seen, like in the example indicated by an arrow in Fig 5A, only in the cell periphery. To estimate the distribution of fluorescence in the cell periphery relative to the cell interior we measured in 30 randomly chosen cells the ratio of mean fluorescence from the entire cell image with ($F_{cell}$) and without ($F_{cyt}$) the peripheral ring. The value of this ratio $F_{cell}/F_{cyt}$ is 1.8 ±0.7,

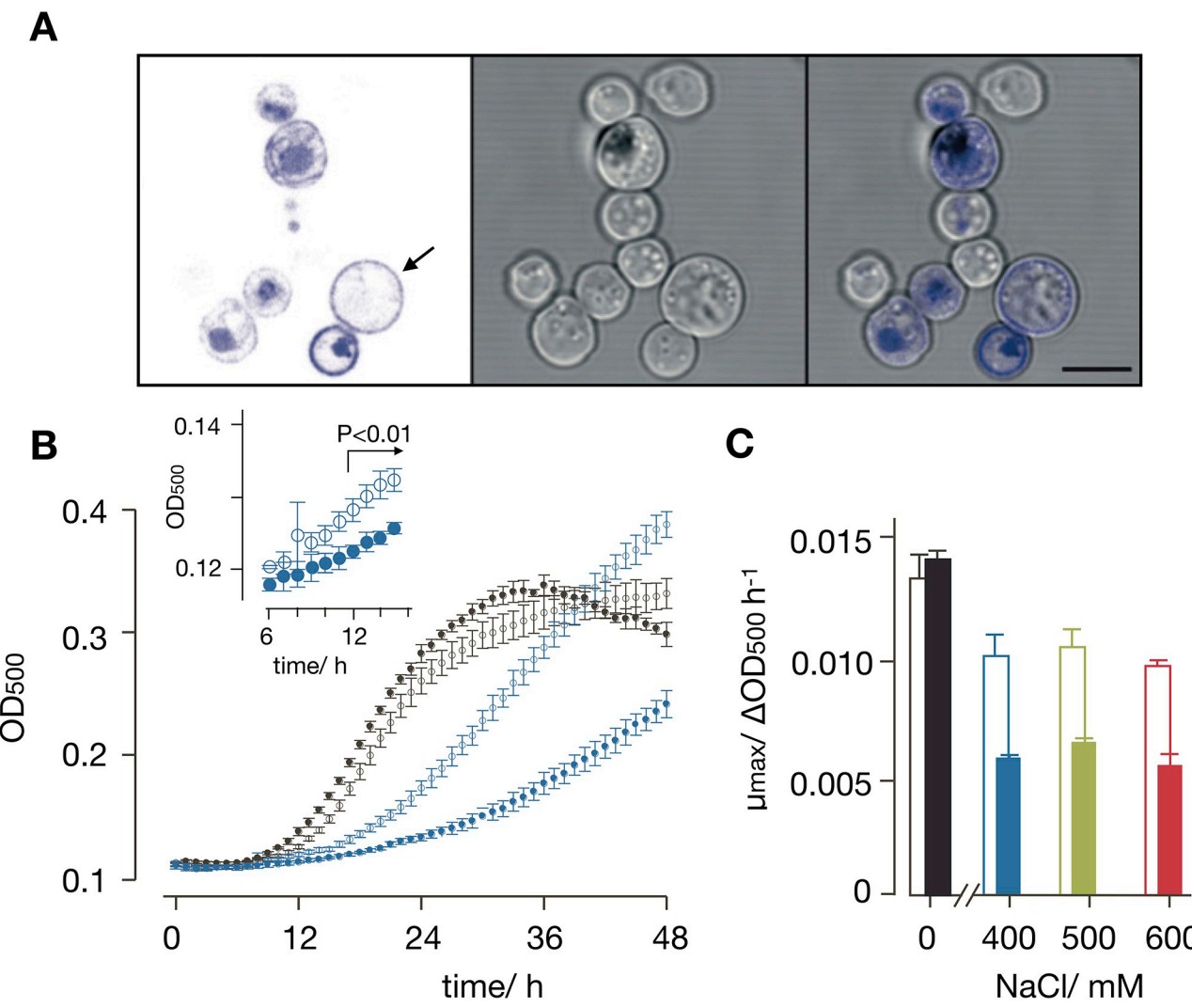

**Fig 5. Expression and functional characterization of N/K-ChR2 in SHY4 cells. (A)** Fluorescence (left) and transmission light (center) images of SHY4 cells constitutively expressing N/K-ChR2-eYFP. Overlay (right) confirms localization of fluorescent tagged channel (in blue) with endo-membranes and the plasma membrane. The cell indicated by arrow exhibits fluorescence nearly exclusively in plasma membrane. Scale bar = 5 μm. **(B)** Growth of SHY4 cells transfected either with empty vector (open symbols) or channelrhodopsin-2 variant (N/K-ChR2, closed symbols) in SD-ura + 10 mM K$^+$ medium with 0 (black) or 400 mM NaCl (blue). Growth measured in 30 sec intervals as change in OD$_{500}$. While the presence of N/K-ChR2 has little impact on growth in absence of NaCl it reduces growth in the presence of salt stress. Inset: data in 400 mM NaCl replotted for cells ± N/K-ChR2. After ≥12 h cells without N/K-ChR2 grow (P<0.01) significantly more than controls. **(C)** Maximum growth rate (μ$_{max}$) from data in B for cells with empty vector (open bars) and with N/K-ChR2 (filled bars) in medium with 0 or 400 mM NaCl. Data are mean ± SD of 3 independent experiments. P values were calculated by student t-test.

confirming that the peripheral ring shows on average a much higher fluorescence signal than the inner part of the cells. Since the tonoplast in yeast cells is well separated from the plasma membrane, the intense fluorescence of the peripheral ring underpins an efficient sorting of the protein to the plasma membrane.

After engineering yeast cells which constitutively produce all-*trans* retinal, we reasoned that ChR2 should be functional in these cells. Given that the light-activated channel is conducting Na$^+$, we repeated Na$^+$ mediated growth inhibition experiments. For that, cells were grown in liquid medium for 48 h in 24 well plates and the optical density at 500 nm (OD$_{500}$) was continuously monitored in a plate reader; one OD$_{500}$ unit is here equivalent to 2.3 x 10$^{-7}$ cells/ml.

The blue-green light from a 20 W tungsten halogen light source served here both as a stimulus for activating ChR2 and as a light source for measuring the $OD_{500}$. Control cells and cells transformed with ChR2 were incubated in SD-ura medium, supplemented with 10 mM KCl plus different concentrations of NaCl (0, 400, 500 or 600 mM NaCl). It should be noted that standard synthetic defined media (SD media) comprise an intrinsic $K^+$ content (from Yeast Nitrogen Base, YNB) of 7 mM. A total concentration of 17 mM $K^+$, as in SD-ura + 10 mM $K^+$, is just sufficient to allow slow growth of these $K^+$ uptake deficient cells (*trk1Δtrk2Δ*-background). We expected that these conditions with limiting $K^+$ supply in the *trk1Δtrk2Δ*-background, resulting in low growth, might be more susceptible to $Na^+$ stress, which has already been observed elsewhere [37].

Growth of the SHY4 yeast strain transformed with N/K-ChR2 was measured in a plate reader in 30 s intervals over time by illuminating cells with 500 nm light. The resulting growth curve shows that cells grow under these conditions in the absence of $Na^+$ (Fig 5B). The growth rates are similar irrespective of whether cells were transformed with the empty vector or N/K-ChR2. Control cells with empty vector start growing in the presence of 400 mM NaCl with a delayed and reduced growth rate (Fig 5B and 5C). Cells expressing N/K-ChR2 exhibit an even longer delay and a significantly reduced maximal growth rate (Fig 5B and 5C). The same results were obtained in experiments with higher $Na^+$ concentrations (500 mM and 600 mM); in all cases the presence of $Na^+$ in the medium caused a reduced maximal growth rate and this effect was strongly augmented in cells transformed with N/K-ChR2 (Fig 5C). This is consistent with a scenario in which N/K-ChR2 is active in the all-*trans* retinal producing cells. Its activation by light elevates the conductance of the plasma membrane for $Na^+$ and the resulting entry of $Na^+$ into the cells is exerting a negative effect on yeast growth. Hence already with this simple growth assay it is possible to test the function of N/K-ChR2 in yeast; after ca. 12h of incubation it is already possible to detect the functional effect of N/K-ChR2 compared to control cells (Fig 5B). In future experiments this system can be further improved by deleting additional genes such as *ENA1-5* [48], *NHA1* [49] and *NHX1* [50], which encode for the main transporters for $Na^+$ export from the cytosol in yeast. This additional modification will make the test system even more sensitive and presumably abolish growth in the presence of high $Na^+$ concentrations.

## $K^+$ complementation

In the next assay we tested whether N/K-ChR2 can accomplish light-dependent $K^+$ mediated complementation of growth in the potassium uptake deficient SHY4 cells. Cells transformed with N/K-ChR2 were therefore grown on agar plates with medium containing high or low $K^+$; the latter condition serves as a selection medium. 10-fold serial dilutions of a cell suspension with $OD_{600}$ = 1.0 were spotted onto SD-ura agar plates supplemented with high $K^+$ (100 mM) or without $K^+$ supplementation. Plates were then incubated under constant blue light illumination (450 nm, 40 μW/mm$^2$) or in the dark. Inspection of cells 72 h after incubation shows that they were growing in high $K^+$ concentration irrespectively on whether they were kept in light or dark. In contrast cells failed to grow on medium with a low $K^+$ concentration independent on whether they were illuminated by blue light or not (S2 Fig). Different from our expectations, these data imply that an activation of N/K-ChR2 by light does not provide a $K^+$ conductance, which is able to rescue growth of the $K^+$ uptake deficient yeast in low $K^+$.

One reason for this negative result could be that the rescue effect of N/K-ChR2 is only modest and not visible in the growth assay on agar. We therefore tested whether we could detect rescue of growth in liquid culture assays as in Fig 5. Growth of SHY4 cells, transformed with either empty vector (ev) or N/K-ChR2 suspended in SDAP-ura medium supplemented with

additional 0.1, 1, 10 or 100 mM KCl were monitored as in Fig 5B in a plate reader. They were therefore illuminated with 500 nm light which served for activating N/K-ChR2 and for measuring changes in OD of the cells. In this condition over 48h, cells grew only in medium with $\geq$10 mM KCl, but without difference between presence/absence of N/K-ChR2 (Fig 6A). The results of these experiments indicate that the plate reader light is sufficient for activating sufficient N/K-ChR2 channels for eliciting a toxic $Na^+$ influx (compare Fig 4B) but not for providing enough $K^+$ influx for growth.

In the next step cells were therefore incubated in wells of a growth plate loaded with SD-ura medium with no (0 mM) or a high (400 mM) concentration of NaCl on a background of a low (7 mM) or medium high (17 mM) KCl concentration. Control cells with empty vector and cells transformed with N/K-ChR2 were continuously illuminated with 465 nm at 40 $\mu W/mm^2$ or kept in the dark. Growth of N/K-ChR2 expressing cells was measured after 48 h as an increase in $OD_{600}$ relative to growth of empty vector controls. The data in Fig 6B show that cells transformed with N/K-ChR2 grow in the dark with and without NaCl in the medium like the respective empty vector controls. In medium with high NaCl, growth in blue light is inhibited by $\geq$35% in the N/K-ChR2 transformed cells irrespective of the KCl concentration (Fig 6B). The results of these experiments are in good agreement with data in Fig 5B and 5C in that activation of N/K-ChR2 promotes $Na^+$ influx, which in turn inhibits cell growth.

Interesting to note is that only cells transformed with N/K-ChR2 and exposed to blue light experience in a selective medium with low (7 mM) KCl and no NaCl a significant stimulation of growth (Fig 6C, middle bars). Here, the final $OD_{600}$ after 48 h is 65% higher than the respective empty vector controls. This finding is consistent with the idea that activation of N/K-ChR2 generates a conductance for $K^+$, which can rescue the deficient endogenous $K^+$ uptake systems in low $K^+$ medium. To further test this assumption the same experiments were repeated in medium with an elevated $K^+$ concentration (17 mM) e.g. a concentration, which is no longer suppressing growth of the yeast mutant. In this scenario the external $K^+$ concentration and therefore $K^+$ influx is already high enough to allow growth of the *trk1Δtrk2Δ*-strain, therefore additional activation of the ChR2 mediated $K^+$ conductance has only little further positive effect on growth (Fig 6C, right bars).

The light dependency of N/K-ChR2 predicts that growth rescue in a selective low $K^+$ medium should be effective in blue but not in red light. To test this prediction, experiments were repeated with either red light or blue light. To address the question whether the stimulating effect of blue light under these conditions is already viable at a shorter incubation time, cell growth was monitored 24h after inoculation. The data in Fig 6D show again no large difference in growth between cells with or without N/K-ChR2 in the dark but a strong growth stimulation in blue light. This underpins that a positive effect of blue light can already be detected after 24h of incubation. In the same assay red light has no perceivable impact on cell growth (Fig 6D). Collectively also these data agree with a scenario in which N/K-ChR2 is active in blue light in the engineered yeast cells and that simple growth assays can be used to test for channel activity.

## Conclusion

The yeast *Saccharomyces cerevisiae* offers many experimental benefits for a fast and easy screening of ion channel proteins [24–30]. Despite of this experimental advantages *S. cerevisiae* has so far not yet been employed for engineering or modifying rhodopsin-based optogenetic tools because yeast neither synthesizes the essential cofactor retinal nor its carotenoid precursor. To overcome this obstacle, we have successfully engineered and optimized a yeast strain, which contains the full metabolic pathway for the synthesis of retinal. The data show that retinal endogenously generated by these yeast cells is sufficient to guarantee robust light dependent

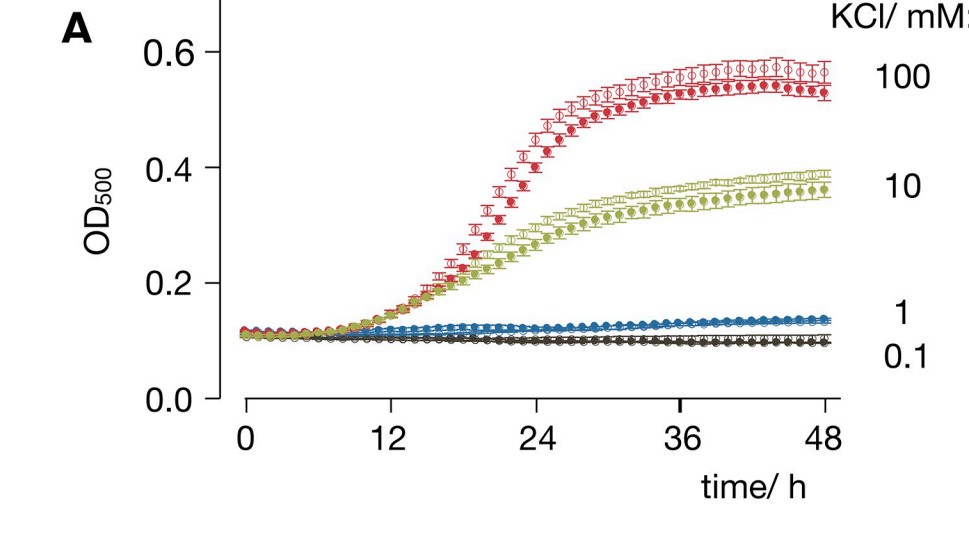

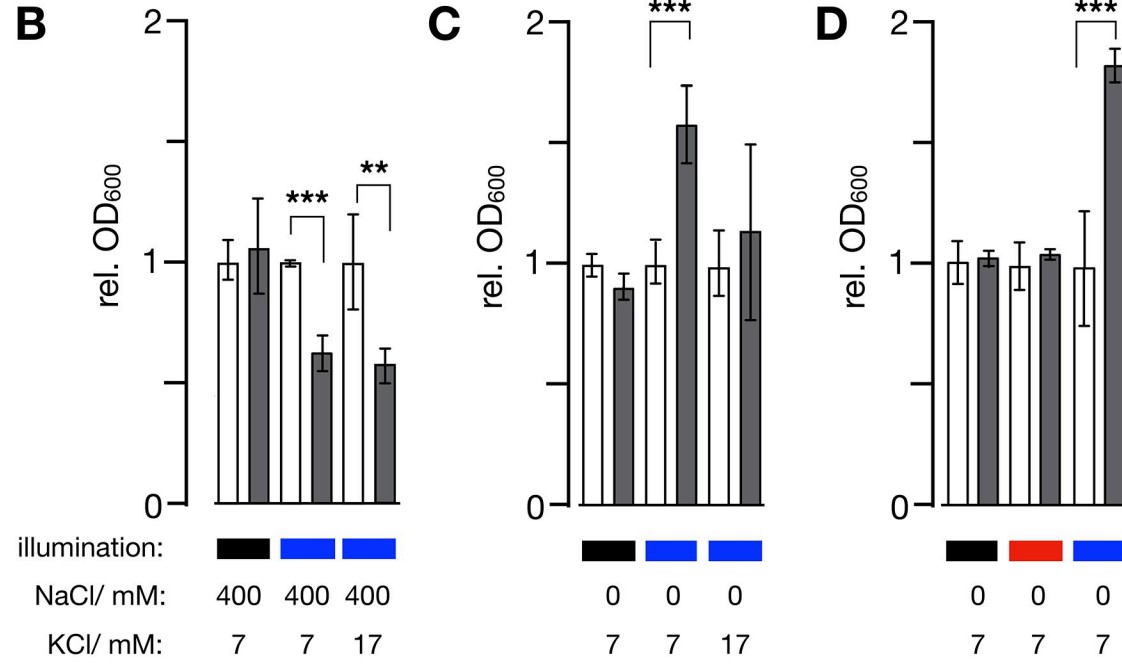

**Fig 6. N/K-ChR2 rescues K$^+$ mediated growth of yeast mutant only in high blue light. (A)** Growth of SHY4 cells transformed with either empty vector (ev, open symbols)) or N/K-ChR2 (closed symbols) in SDAP medium with 0.1 (black), 1 (blue), 10 (green) or 100 (red) mM KCl. Growth was monitored as in Fig 5B. When cells were illuminated by 500 nm light from the plate-reader, they grew over a period of 48h only in medium with ≥10 mM KCl but without appreciable difference between presence/absence of N/ K-ChR2. Data are mean ± SD of 3 independent experiments. **(B)** Relative growth of SHY4 cells expressing N/K-ChR2 in dark (black bar) or illuminated with 40 μW/mm² of light (465 nm: blue bar) in SD-ura medium ± 400 mM NaCl on a background of low (7 mM) or medium high (17 mM) KCl. Growth expressed as OD value (rel. OD$_{600}$) of N/K-ChR2 expressing cell (grey bar) relative to the respective control with empty vector (open bars). Growth is substantially reduced by NaCl only in cells expressing N/K-ChR2 upon light exposure. **(C)** Growth is increased in blue light grown cells expressing N/K-ChR2 only in low K$^+$ medium e.g. when K$^+$ influx is the limiting growth factor, but not in medium high K$^+$ (17 mM). **(D)** In low K$^+$ (7 mM), growth of cells expressing N/K-ChR2 is increased in cells exposed to blue light (blue bar), but not in cells exposed to red light (650 nm, red bar) or kept in the dark (black bar). Data in **(B)** and **(C)** represent growth after 48 h, data in **(D)** represent growth after 24 h. All data are mean ± SD of ≥3 independent experiments. P value of statistical significance from t-test indicated as * = P<0.05, ** = P<0.01 and *** = P<0.001.

activity of ChR2 in *S. cerevisiae*. In proof of concept experiments we confirm that the weak cation selective conductance of this channel [10], rescues in the retinal producing strain, growth of a $K^+$ uptake-deficient yeast mutant or inhibits growth by high extracellular $Na^+$ in a light dependent manner. Collectively, the simple experiments already show that the retinal producing yeasts can be used as a reliable and cheap test system for screening basic ChR2 functions. Given the simplicity of the system and the robust nature of optical density as a read-out signal, the system can be easily scaled up in future for high-throughput screening including robotics platforms. Promising candidate proteins, which emerge from this screening of large libraries, can in the next step be characterized in depth with conventional electrophysiological assays. A combination of the present screening systems with a bias for $K^+$ uptake to promote growth and a bias against $Na^+$ influx to prevent growth inhibition is only one example for an experimental strategy to screen for mutant ChR2 channels with increased $K^+$ over $Na^+$ selectivity. By combining the benefits of genetic modifications in yeast with the wealth of yeast mutants with defined functional phenotypes it will be possible to create a versatile platform for engineering ChR2 and related retinal dependent proteins like baceriorhodopsins with new channel functions.

## Materials and methods

### Generation of plasmids

For all cloning steps the NEBuilder HiFi DNA Assembly Master Mix (NEB GmbH, Frankfurt, Germany) was used. Generated plasmids in Table 1 were amplified in *E. coli* DH5α and sequence verified by Sanger sequencing (Microsynth Seqlab GmbH, Göttingen, Germany).

Plasmid pGREG506-P$_{TEF1}$-ChR2-eYFP was generated by cloning PCR amplified ChR2-eYFP (P19/P20) using pGEM-HE-ChR2-eYFP [10] as template in *Sal*I restricted pGREG506-P$_{TEF1}$. Plasmid pGREG506-P$_{TEF1}$-N/K-ChR2-eYFP was generated by cloning PCR amplified N/K-ChR2-eYFP (P21/P22) using pGEM-N/K-ChR-eYFP as template in *Sal*I restricted pGREG506-P$_{TEF1}$.

Plasmid pGREG576-TRPV1 was generated by cloning PCR amplified TRPV1 (P23/P24) using pcDNA3.1-TRPV1 [54] as template in SalI restricted pGREG576.

CRISPR/Cas9 plasmids were generated by using KpnI and PmeI restricted pCAS9c [43] with corresponding gRNA primers for plasmid assembling with the NEBuilder: P3/P4 for pCas9c-*CAN1*, P9/P10 for pCas9c-k*an*$^r$ and P15/P16 for pCAS9c-YPRCΔ15. The protospacer sequences were designed by using the ATUM gRNA Design Tool (https://www.atum.bio/eCommerce/cas9/input).

The polycistronic β-carotene biosynthetic pathway (P$_{GAP}$-crtYB-T2A-crtI-T2A-crtE-T$_{TEF1}$) was PCR amplified (P1/P2) from pUDC082 as template. carX was PCR amplified (P7/P8) from pGEX-carX as template. crtI was PCR amplified (P13/P14) from YIplac211-YB/I/E as template.

Colony PCRs were performed with OneTaq 2X Master Mix in Standard Buffer (NEB) according to manufacturer's instructions with following modifications: To prepare DNA template, cells from a single colony were suspended in 30 μl of 0.02% SDS, incubated 10 min at 95˚C and vortexed vigorously for 1 min. 0.5 μl of the cell suspension was used as template for the PCR reaction with corresponding colony PCR primers.

pUDC082 was a gift from Jules Beekwilder [41], pGEX-carX from Salim Al-Babili [43], and YIplac211-YB/I/E from Gerhard Sandmann [40], and pcDNA3.1-TRPV1 from Makoto Tominaga [54].

### Yeast strains

*S. cerevisiae* strains used in this study are listed in Table 2.

**Table 1. Plasmids used in this study.**

| name | properties | reference |
|---|---|---|
| pGREG506 | empty vector with *GAL1* promoter in front of *HIS3* stuffer | [51] |
| pGREG506-P$_{TEF1}$ | pGREG506 with inducible *GAL1* promoter replaced by constitutive *TEF1* promoter | [52] |
| pGREG506-ChR2 | pGREG506 with *HIS3* stuffer replaced by ChR2 | This study |
| pGREG506-D156C | pGREG506 with *HIS3* stuffer replaced by D156C mutant of ChR2 | This study |
| pGREG506-ChR2-eYFP | pGREG506 with *HIS3* stuffer replaced by C-terminal eYFP-tagged ChR2 | This study |
| pGREG506-P$_{TEF1}$-ChR2-eYFP | pGREG506-PTEF1 with *HIS3* stuffer replaced by C-terminal eYFP-tagged ChR2 | This study |
| pGREG506-N/K-ChR2-eYFP | pGREG506 with *HIS3* stuffer replaced by C-terminal eYFP-tagged N/K-ChR2 | This study |
| pGREG506-P$_{TEF1}$-N/K-ChR2-eYFP | pGREG506-P$_{TEF1}$ with HIS3 stuffer replaced by C-terminal eYFP-tagged N/K-ChR2 | This study |
| pGREG576 | empty vector with *GAL1* promoter in front of HIS3 stuffer for expression of N-terminal GFP fusion proteins | [51] |
| pGREG576-TRPV1 | pGREG576 with *HIS3* stuffer replaced by TRPV1, expression of N-terminal TRPV1-GFP fusion protein under control of GAL1 | This study |
| pEVP11-AEQ | 2μ-plasmid containing *LEU2* marker and the gene for constitutive expression of apoaequorin under control of the ADH-promoter | [53] |
| pCAS9c | all-in-one CRISPR/Cas9 plasmid with gRNA expression cassette containing protospacer stuffer | [42] |
| pCAS9c-*CAN1* | pCAS9c with: gRNA expression cassette containing sprotospacer sequence for *CAN1* | This study |
| pCAS9c-kan$^r$ | pCAS9c with: gRNA expression cassette containing protospacer sequence for kan$^r$ | This study |
| pCAS9c-YPRCΔ15 | pCAS9c with: gRNA expression cassette containing protospacer sequence for YPRCΔ15 | This study |
| pUDC082 | template for PCR amplification of P$_{GAP}$-crtYB-T2A-crtI-T2A-crtE-T$_{TEF1}$ | [41] |
| pGEX-carX | template for PCR amplification of carX | [43] |
| YIplac211-YB/I/E | template for PCR amplification of crtI | [40] |
| pcDNA3.1-TRPV1 | Template for PCR amplification of TRPV1 | [54] |

The table specifies the names and essential properties of the plasmids used in this study.

## Engineering of an all-trans retinal producing *trk1Δtrk2Δ* yeast strain

In order to generate a βproducing yeast strain, the K$^+$-uptake deficient strain PLY240 was co-transformed with pCas9c-*CAN1* and the PCR amplified polycistronic β-carotene biosynthetic pathway (P$_{GAP}$-crtYB-T2A-crtI-T2A-crtE-T$_{TEF}$). Transformed cells were spread on SD-ura

**Table 2. Yeast cells used in this study.**

| name | genotype | reference |
|---|---|---|
| BY4741 | *MATa his3Δ1 leu2Δ0 met15Δ0 ura3Δ0* | [55] |
| PLY240 | *MATa; his3Δ200; leu2-3,112; trp1Δ901; ura3-52; suc2Δ9; trk1Δ51; trk2Δ50::loxP-kanMX-loxP* | [37] |
| SHY1 | PLY240; *can1*::P$_{GAP}$-crtYB-T2A-crtI-T2A-crtE-T$_{TEF1}$ | This study |
| SHY2 | SHY1; *trk2Δ50::loxP*-P$_{TEF1}$-*kan$^r$*::carX-T$_{TEF1}$-*loxP* | This study |
| SHY3 | SHY1; YPRCΔ15::P$_{GAP}$-crtI-TCYC1 | This study |
| SHY4 | SHY3; *trk2Δ50::loxP*-P$_{TEF1}$-kanr::carX-T$_{TEF1}$-*loxP* | This study |

All yeast strains used in this study and corresponding genotypes are listed in this table.

medium +100 mM KCl and grown for 3 days to select for the presence of the pCas9c-*CAN1*-plasmid. Colonies showing yellow color indicated production of β-carotene and therefore presence of the polycistronic β-carotene pathway. Randomly selected yellow colonies were analyzed by colony PCR (P5/P6) for correct integration of the polycistronic β-carotene biosynthetic pathway in the *CAN1* locus. The PLY240 strain containing the correctly integrated polycistronic β-carotene biosynthetic pathway in the *CAN1* locus was dubbed SHY1.

SHY1 cells were co-transformed with pCas9c-kan$^r$ and the PCR amplified carX to generate SHY2. Transformed cells were selected on SD-ura medium +100 mM KCl. Decrease in intensity of yellow color indicated genomic integration of carX. Randomly selected colorless cells were analyzed by colony PCR (P11/P12) for correct integration of carX in the kan$^r$ locus.

SHY1 cells were co-transformed with pCas9c-*YPRCΔ15* and the PCR amplified crtI to generate SHY3. Transformed cells were selected on SD-ura medium +100 mM KCl. Coloring of cells indicated genomic integration of crtI. Randomly selected orange colonies were analyzed by colony PCR (P17/P18) for correct localized integration of crtI in the *YPRCΔ1* locus.

SHY4 was generated by co-transforming SHY3 cells with pCas9c-kan$^r$ and the PCR amplified carX. Transformed cells were selected on SD-ura medium +100 mM KCl. Colorless cells indicated genomic integration of carX. Randomly selected colorless cells were analyzed by colony PCR (P11/P12) for correct integration of carX in the kan$^r$ locus.

In order to remove the pCAS9c plasmids from successfully transformed and positively PCR-tested cells, the respective cells were transferred to complex YPD medium + 100 mM KCl and subjected to three serial overnight passages. From the final overnight culture, 100–200 cells (1:10.000 dilution from OD$_{600}$ = 1.0) were plated on YPD + 100 mM KCl agar plates and grown for 2 days. Colonies were picked and streaked on SD-ura + 100 mM KCl for counter selection. Cells which did grow on YPD + 100 mM KCl, but not on SD-ura + 100 mM KCl were plasmid free and used for further work.

## All-trans retinal extraction

For all-*trans*-retinal extraction a freshly grown overnight culture (YPD +100 mM KCl) of cells was harvested by centrifugation (5 min, 4000 rpm, 4˚C). The pellet was washed twice with ice cold 5 mL ddH$_2$O and the supernatant was removed. The pellet was resuspended in 1 mL 100% EtOH and transferred in a 1.5 mL reaction tube. 500 µl of 0.5 mm Zirconia/glass beads (Carl Roth) were added and the cells were homogenized three times for 60 sec on a Minilys® Homogenizer (Bertin Technologies). After each round of bead beating the suspension was cooled on ice for 60 sec. The suspension was centrifuged (1 min, 13000 rpm, 4˚C) and the supernatant containing the carotenoids was transferred to a vial for UPLC analysis. The vial was anaerobized using 4 times alternating degassing by a vacuum pump and gassing with nitrogen.

## UPLC analysis

UPLC analysis of the ethanol cell extract from SHY4 cells with 5 µM all-*trans*-retinal as reference was kindly performed by Dr. Markus Krischke at the University of Würzburg as described elsewhere [10].

## Yeast media

The host strains BY4741 and PLY240 were routinely cultured in YPD (2% glucose; 2% peptone; 1% yeast extract), which was supplemented by 100mM KCl for PLY240.

Strains harboring plasmids with the *URA3* marker were selected and grown in SD-ura (2% glucose; 0.69% yeast nitrogen base without amino acids; 0.077% CSM Drop-out -Ura). Agar

media contained 2% agar-agar. $K^+$ and $Na^+$ concentrations were adjusted with KCl or NaCl to the indicated concentrations.

For growth tests in media containing $K^+$ concentrations below 7 mM (intrinsic level of standard SD media) Synthetic Defined Arginine Phosphate (SDAP) media [56, 57] was used, which is composed of 10 mM arginine, 2% glucose, 2 mM $MgSO_4$, 0.2 mM $CaCl_2$, 1% trace elements (13 µM FeNaEDTA, 8 µM $H_3BO_3$, 0.25 µM $CuSO_4$, 0.6 µM KI, 2.7 µM $MnSO_4$, 1 µM $Na_2MoO_4$, 2.5 µM $ZnSO_4$, 0.5 µM $CoCl_2$, 0.5 µM $NiCl_2$), 1% vitamins (2 µg/l biotin, 400 µg/l Ca-panthotenate, 2 µg/l folic acid, 200 µg/l inositol, 400 µg/l niacin, 200 µg/l p-aminobenzoic acid, 400 µg/l pyridoxine hydrochloride, 200 µg/l riboflavin, 400 µg/l thiamine hydrochloride), 50 mg/l His, 50 mg/l Trp, 50 mg/l Leu. The required $K^+$ concentration was added as KCl and pH was adjusted to 5.5 with 85% $H_3PO_4$.

## Oligonucleotides and plasmids

All oligonucleotides used are listed in Table 3. PCR was performed by using the Q5$^®$ Hot Start High-Fidelity 2X Master Mix (NEB GmbH, Frankfurt, Germany) according to manufacturer's instructions. Plasmids employed and generated in this study are listed in Table 1.

For all cloning steps the NEBuilder HiFi DNA Assembly Master Mix (NEB GmbH, Frankfurt, Germany) was used. Generated plasmids were amplified in *E. coli* DH5α and sequence verified by Sanger sequencing (Microsynth SeqlabGmbH, Göttingen, Germany).

## Yeast media

Strain PLY240 was routinely cultured in YPD + 100 mM KCl (2% glucose; 2% peptone; 1% yeast extract; 100 mM KCl). PLY240 strains harboring plasmids were selected and grown in corresponding synthetic defined (SD) drop-out media (2% glucose; 0.69% yeast nitrogen base without amino acids; 0.077% CSM Drop-out -Ura or 0.067% CSM Dropout -Leu-Ura or 0.062% CSM Drop-out-Leu-Trp-Ura). Agar media contained 2% agar (AppliChem, Darmstadt, Germany).

## Yeast transformation

Transformation of yeast strain PLY240 with the generated plasmids was performed by using the Frozen-EZ Yeast Transformation II kit (Zymo Research Europe GmbH, Freiburg, Germany) according to manufacturer's instructions. For selection, transformed yeast were streaked onto SD-ura +100 mM KCl agar plates, incubated at 30˚C and colonies were picked after 72 h.

## Luminometric assay

Aequorin-based luminometry was used to monitor changes in intracellular calcium concentration in intact yeast cells. For that, cells were grown overnight in the dark in 10 mL SGal-leu medium containing 3 µL of 6.8 mM coelenterazine. Cells were harvested by centrifugation, resuspended in an incubation medium (10 mM Tris/MES, 0.1 mM $CaCl_2$, 10 g/L glucose at pH 7) to an $OD_{600}$ = 2 and incubated for 30 min at 30˚C in the dark. A 100 µL aliquot of the cell suspension was loaded into the luminometer (Lumac/3M, Abimed). 10 s after start of the recording, 10 µM capsaicin was added by injecting 100 µL of test medium (same as incubation medium plus 20 µM capsaicin) into the cuvette. Data acquisition was done with Labview Signal Express (National Instruments) and a NI USB-6009 interface (National Instruments).

**Table 3. Oligonucleotides used in this study.**

| name | sequence (5'- 3') | amplicon |
|---|---|---|
| P1 (fw) | TCTGTACTTCTCCTTCATCTTCATCACCTATGCCAATCCTTTGCCGTAGTTTCAACGTATG | polycistronic β-carotene pathway: P_GAP-crtYB-T2A-crtI-T2A-crtE-T_TEF1 |
| P2 (rv) | CCGACGAGAGTAAATGGCGAGGATACGTTCTCTATGCCAGTATAGCGACCAGCATTCACATACGATTGACG | polycistronic β-carotene pathway: P_GAP-crtYB-T2A-crtI-T2A-crtE-T_TEF1 |
| P3 (fw) | GATACGTTCTCTATGGAGGAGTTTTAGAGCTAGAAATAGCAAGTTAAAATAAG | gRNA-CAN1 |
| P4 (rv) | TCCTCCATAGAGAACGTATCGATCATTTATCTTTCACTGCGGAG | gRNA-CAN1 |
| P5 (fw) | GAGATAGATACATGCGTGGGTC | colony PCR: polycistronic β-carotene pathway in CAN1 |
| P6 (rv) | GATGCCACGGTATTTCAAAG | colony PCR: polycistronic β-carotene pathway in CAN1 |
| P7 (fw) | CACATCACATCCGAACATAAACAACCATGAGAGTTCTGCAACAAAATTCCTTCACACAAACG | carX |
| P8 (rv) | CAAGAATCTTTTTATTGTCAGTACTGATTATCATCCAACAGCTTTCTCCAACTTCTCTCG | carX |
| P9 (fw) | TTACTCACCACTGCGATCCCGTTTTAGAGCTAGAAATAGCAAGTTAAAATAAG | gRNA-kan^r |
| P10 (rv) | GGGATCGCAGTGGTGAGTAAGATCATTTATCTTTCACTGCGGAG | gRNA-kan^r |
| P11 (fw) | CATCTGGGCAGATGATGTCG | colony PCR: carX in kan^r |
| P12 (rv) | GTGTCGGATGACTTCTTCTACG | colony PCR: carX in kan^r |
| P13 (fw) | AGAAAGAAAAACTAACACATTAATGTAGTTTTAAAATTTCAAATCCGAACAACAGAGCATAGGGTTTCGCAAAAGAGGATCCCCGGGTACCCAGTTCGAG | P_GAP-crtI-T_CYC1 |
| P14 (rv) | TAGCACAAATAATACCGTGTAGAGTTCTGTATTGTTCTTCTTAGTGCTTGTATATGCTCATCCCGACCTTCCATTCTGGAATTCGAGCTCGGTACCGGCCG | P_GAP-crtI-T_CYC1 |
| P15 (fw) | ATATGTTTGGTTTCGATTGTGTTTTAGAGCTAGAAATAGCAAGTTAAAATAAG | gRNA-YPRCΔ15 |
| P16 (rv) | ACAATCGAAACCAAACATATGATCATTTATCTTTCACTGCGGAG | gRNA-YPRCΔ15 |
| P17 (fw) | CAAGTTCTTGGTTTCAGGCC | colony PCR: crtI in YPRCΔ15 |
| P18 (rv) | ATAAAGCAGCCGCTACCAAA | colony PCR: crtI in YPRCΔ15 |
| P19 (fw) | GAATTCGATATCAAGCTTATCGATACCGTCGACAATGGATTATGGAGGCGCCCTG | ChR2-eYFP |
| P20 (rv) | GCCGTGACATAACTAATTACATGACTCGAGGTCGACTTACTTGTACAGCTCGTCCATGCC | ChR2-eYFP |
| P21 (fw) | GAATTCGATATCAAGCTTATCGATACCGTCGACAATGCGACCCCAAATACTCCTCTTG | N/K-ChR2-eYFP |

(Continued)

**Table 3.** (Continued)

| name | sequence (5'- 3') | amplicon |
|---|---|---|
| P22 (rv) | GCGTGACATAACTAATTACATGACTCGAGGTCGACTTAAACTTCATTTTCATAGCAAAATCTAGACTTGTACAGCTCGTC | N/K-ChR2-eYFP |
| P23 (fw) | GAATTCGATATCAAGCTTATCGATACCGTCGACAATGGAACAACGGGCTAGCTTTA | GFP-TRPV1 |
| P24 (rv) | GCGTGACATAACTAATTACATGACTCGAGGTCGACTTATTTCTCCCCTGGGACCA | GFP-TRPV1 |

The underlined parts of the sequence mark homology regions of the overhang primers for NEBuilder-based cloning.

## Growth assays

All strains were grown overnight in appropriate SD + 100 mM KCl medium. Grown cells were washed twice with ddH$_2$O and set to OD$_{600}$ = 1.0. 10-fold dilutions were spotted on corresponding SD agar plates + 1 mM KCl and as a control on plates with + 100 mM KCl. Plates were incubated at 30˚C for 72 h and illuminated by an 8W Albrillo LED Daylight Lamp with constant 300 μEinstein or placed in a sealed box to avoid any illumination. This device consists of a square, 5 mm thick aluminum base plate with a side length of 300 mm to deflect the heat generated by 12 WINGER® WEPBL3-S1 Power (465nm) 3W LEDs, which are centered at the appropriate spacing for wells of a 24 well plate. This device, called "light plate", serves as base for a "growth plate". On this growth plate, 12 wells are illuminated at a time and 12 remain non-illuminated. The illumination intensity can be modulated by the applied voltage to the LEDs or the distance between them and the "light plate". Since the use of the "light plate" did not allow continuous tracking of cell growth, the OD$_{600}$ was instead measured manually at regular intervals with a plate reader at on a BioTek EpochTM 2 Microplate Spectrophotometer (Aligen Santa Clara, Ca USA).

In an alternative approach cells were incubated as described above in 24 well plates and the latter were incubated for 48 h at a constant temperature of 30˚C in a BioTek EpochTM 2 Microplate Spectrophotometer (Aligen Santa Clara, Ca USA). The well plates, sealed with Breathe-Easy sealing membranes (Merck, Darmstadt, Germany), were illuminated in 30 sec intervals with blue light of 500 nm. The latter served both to activate the ChR variant of interest and to monitor the OD$_{500}$.

To determine the maximum growth rate (μ$_{max}$), the growth curves were first smoothed by calculating the moving average (window width = 5 data points). Subsequently, the growth rate μ for two neighboring data points was determined by Eq 1 calculating the difference quotient

$$\mu\left(t + \frac{\Delta t}{2}\right) = \frac{OD_{500}(t + \Delta t) - OD_{500}(t)}{\Delta t} \tag{1}$$

with t being the time and Δt the time between two neighboring data points. The highest value was then used as μ$_{max}$.

## Confocal microscopy

Cells expressing eGFP or eYFP tagged proteins were imaged on a Leica TCS SP5 II spectral confocal laser scanning microscope (Leica Microsystems, Heidelberg Germany) equipped with a 100-x oil objective (HCX PL APO CS 100x/1.44). GFP/YFP was excited at 488 nm and emission was detected between 510–585 nm.

The intensity of yellow color in Fig 2 was analyze with Fiji imaging software [58] using the *color deconvolution* plugin tool. Regions of interest on the agar plates and on the background were unmixed into three distinct colors of which one was yellow. The intensity in the yellow channel from a region of interest in the background was subtracted from the respective area over the agar plates.

## Plasmids for electrophysiology

The ChRs in the pGEMHE vector was from the stock of the Nagel lab. Point mutations were introduced by QuikChange PCR. To remove the original plasmid, the PCR mix after amplification was incubated with DpnI enzyme for 3 hours at 37˚C. All constructs were verified by automated DNA sequencing.

## Electrophysiology in oocytes

Oocytes *Xenopus laevis* were obtained from the Department of Molecular Plantphysiology and Biophysics—Botany I of the University Wuerzburg. The laparotomy to obtain oocytes from *Xenopus* was carried out in accordance with the principles of the Basel Declaration and recommendations of Landratsamt Wuerzburg Veterinaeramt. The protocol under License #70/14 from Landratsamt Wuerzburg, Veterinaeramt, was approved in written form by the responsible veterinarian.

*X. laevis* oocytes were injected with 30 ng cRNA and incubated in medium containing 10 µM all-*trans*-retinal for 2 or 3 days before measurement. Two-electrode voltage-clamp recordings of photo-currents were made in Ringer's solution (110 mM NaCl, 5 mM KCl, 2 mM $BaCl_2$, 1 mM $MgCl_2$, 5 mM HEPES, pH 7.6) or in solutions with high concentration of $Na^+$ or $K^+$ (120 mM NaCl or KCl, 2 $BaCl_2$, 5 mM HEPES, pH 7.6) or in solutions with low concentration of $Na^+$/$K^+$ (1 mM NaCl or KCl, 119 NMG, 2 $BaCl_2$, 5 mM HEPES, pH 7.6). Electrode capillaries were filled with 3 M KCl. Stimulation and data acquisition were controlled with an AD-DA converter (Digidata 1322A, Axon Instruments) and WinWCP software (Strathclyde University, United Kingdom). A 473 nm laser (Changchun New Industries Optoelectronics Tech) were used as light sources.

## Statistics

Data form yeast growth assays in liquid culture (Figs 5B, 5C and 6A–6D) and electrophysiological recordings of photocurrents (Fig 4B and 4E) are mean values ± standard deviation from ≥ 3 independent biological replicates. All images of yeast growth experiments on agar plats (Fig 1B, 1D, S1 and S2B Figs), images of yeast cells (Figs 1A, 2A, 3A, 3B and 5A), *Xenopus laevis* oocytes (4C), HPLC spectra (Fig 3C) and electrical/ fluorescent trajectories (figs 1C and 4D) are representative examples from ≥ 3 independent biological replicates. Statistical significance was calculated with student t-test.

## Supporting information

**S1 Fig. Functional complementation of SHY4 cells by $K^+$ channel. (A)** Fluorescence images of SHY4 cells transiently expressing GFP tagged $K^+$ channel KcvPBCV1 (KcvPBCV1::GFP). Scale bar = 5 µm. **(B)** Serial dilutions of $K^+$ uptake deficient SHY4 cells transformed with KcvPBCV1::GFP plasmid or the corresponding empty vector (ev). Cells spotted on SD-ura plates with either high (100 mM KCl, top) or without $K^+$ supplementation and incubated for 72h. KcvPBCV1 rescues growth of SHY4 cells on SD-ura medium without additional $K^+$ (bottom row). For experimental details see [1]. [1] Gebhardt M, Hoffgaard F, Hamacher K, Kast SM, Moroni A, Thiel G. Membrane anchoring and interaction between transmembrane domains is crucial for $K^+$ channel function. J. Biol. Chem. 2011; 286:11299–11306.
(PDF)

**S2 Fig. N/K-ChR2-eYFP expression does not restore growth of SHY4 cells on low $K^+$ agar medium.** Serial dilutions of all-*trans* retinal producing $K^+$ uptake deficient SHY4 cells transformed with either a plasmid for constitutive expression of N/K-ChR2-eYFP or the corresponding empty vector (ev). Cells spotted on SD-ura plates with either high (100 mM KCl, top) or a without $K^+$ supplementation and incubated for 72h. Colonies grow only on SD-ura medium with high $K^+$ concentration independently of incubation in the dark (left) or in blue light (right).
(PDF)

**S1 Data.**
(XLSX)

## Acknowledgments

We would like to thank Markus Krischke (Uni. of Würzburg) for HPLC analysis and Tobias Meckel (TU Darmstadt) for help with image analysis.

## Author Contributions

**Conceptualization:** Sebastian Höler, Daniel Degreif, Shiqiang Gao, Georg Nagel, Anna Moroni, Gerhard Thiel, Adam Bertl, Oliver Rauh.

**Data curation:** Sebastian Höler, Florentine Stix, Shang Yang, Gerhard Thiel, Adam Bertl, Oliver Rauh.

**Formal analysis:** Sebastian Höler, Shang Yang, Oliver Rauh.

**Funding acquisition:** Georg Nagel, Anna Moroni, Gerhard Thiel.

**Investigation:** Sebastian Höler, Daniel Degreif, Florentine Stix, Shang Yang, Shiqiang Gao.

**Methodology:** Daniel Degreif, Shiqiang Gao, Anna Moroni.

**Project administration:** Georg Nagel, Anna Moroni, Adam Bertl.

**Supervision:** Daniel Degreif, Shiqiang Gao, Gerhard Thiel.

**Writing – original draft:** Sebastian Höler, Shang Yang, Georg Nagel, Anna Moroni, Gerhard Thiel, Adam Bertl, Oliver Rauh.

**Writing – review & editing:** Gerhard Thiel, Adam Bertl, Oliver Rauh.

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
