## [Decision Letter · Decision Letter 0]

20 Feb 2023

PONE-D-22-35053Tailoring baker’s yeast Saccharomyces cerevisiae for functional testing of ChannelrhodopsinPLOS ONE

Dear Dr. Thiel,

Thank you for submitting your manuscript to PLOS ONE. After careful consideration, we feel that it has merit but does not fully meet PLOS ONE’s publication criteria as it currently stands. Therefore, we invite you to submit a revised version of the manuscript that addresses the points raised during the review process.

We look forward to receiving your revised manuscript.

Kind regards,

Patrick Lajoie, PhD

Academic Editor

PLOS ONE

Journal Requirements:

    "This work was funded in part by the European Research Council (ERC; 2015 Advanced Grant 495 (AdG) n. 695078 noMAGIC (G.T. & A.M.) the priority program 

SPP1926 (to GT) and by Projects #374031971, TRR 240 A04 and #417451587 (to G.N.) from DFG, German Research Foundation. We would like to thank Markus Krischke (Uni. of Würzburg) for HPLC analysis and Tobias Meckel (TU Darmstadt) for help with image analysis."

  "GT; AM: European Research Council

GT:German Research Foundation.

GN:  German Research Foundation.

Additional Editor Comments:

The reviewers have raised several points that I believe should be addressed prior acceptance for publication. In particular, while I think asking for a full screen would be a bit too ambitious for publication in PLoS One, I believe the reviewer as a valid point and further characterization of the function of ChR2 in yeast to highlight the power of the model would be beneficial for the readers so they can appreciate this new tool. 

Reviewers' comments:

Reviewer's Responses to Questions

**Comments to the Author**

1. Is the manuscript technically sound, and do the data support the conclusions?

Reviewer #1: Partly

Reviewer #2: Partly

2. Has the statistical analysis been performed appropriately and rigorously? 

Reviewer #1: No

Reviewer #2: I Don't Know

3. Have the authors made all data underlying the findings in their manuscript fully available?

Reviewer #1: No

Reviewer #2: No

4. Is the manuscript presented in an intelligible fashion and written in standard English?

Reviewer #1: Yes

Reviewer #2: Yes

5. Review Comments to the Author

Reviewer #1: The manuscript by Höller et el., aims to implement the light-activated membrane channel (Channelrhodopsin) in the budding yeast S. cerevisiae. This is a challenging endeavor since yeast is not producing the chromophore (retinal) enabling Channelrhodopsin function. The authors overcome this problem by engineering the metabolic pathway for retinal biosynthesis in yeast. Finally, functionality of Channelrhodopsin was tested indirectly by measuring the yeast growth under different Na+ and K+ concentrations.

Major comments:

- The functionality of the Channelrhodopsin in yeast was tested using a growth assay under different Na+ and K+ concentrations. These indirect results should be complemented with a direct measurement of channel functionality upon light stimulation.

- The growth curves shown in Figures 5 and 7 were performed using light at 500 nm. The authors mention that this wavelength was used to measure the yeast growth and activate the Channelrhodopsin. For yeast growth curves, measurements must be carried out at 600 nm. Therefore, the authors should use an illumination system coupled to a plate reader to stimulate the yeast cells at 460 nm and read the growth at 600 nm. Please see the illumination systems in the following references PMIDs: 27805047, 31788779, 34431694, and 36457859.

- In Figure 5, the constant darkness condition (control condition) is missing.

- Figure 6 should be moved to supplementary information.

- In Figure 7 (panels B, C, and D), the statistical analysis should be done between constant darkness (control) and blue-light.

- In Figure 7 (panel A), the constant darkness condition (control condition) is missing.

Minor comments:

- In table 1, replace “stuffer” by locus.

Reviewer #2: REVIEWER COMMENTS

In this manuscript, the authors report an engineered and validated Saccharomyces cerevisiae strain that expressed the light-activated Channelrhodopsin 2 (ChR2) that could be used for future screens.

Overall, the authors approach to generate a functional light-activated ion transporter using synthetic biology is interesting. From the abstract, I was looking forward to understanding the proposed engineered yeast system and to see what the potential is. However, the manuscript felt very short to provide a toolbox that might be useful to the research community. I recommend the authors to remove some of the distracting data, to add a substantial amount of data to either (1) perform a screen using their engineered yeast or (2) to characterize ChR2 in yeast. I also recommend the authors to polish the manuscript to make it easier to read.

Major points to address

(1) The first two figures contained failed attempt to expressed ChR2 in yeast, so the authors used a different approach in Fig. 3. Fig. 1 and 2 should be removed to the manuscript as they are distracting and not informative nor helpful for the rest of the data.

(2) The authors stated that N/K-ChR2-eYFP localizes at the plasma membrane (PM) with a subpopulation in endomembrane. However, it seems that the majority of ChR2 is at the vacuole with a small fraction at the PM (Fig. 5A). It will suggest that most of ChR2 is non-functional nor folded properly. The authors should further characterize the expression and localization of ChR2 in yeast. ChR2 should mostly localize at the PM. Therefore, I disagree with the authors statement “When the optimized ChR2-5x, termed N/K-ChR2, was expressed in SHY4 cells, we found that it was also in this organism efficiently sorted to the plasma membrane”.

(3) From Fig. 1 title, the authors stated that ChR2 is properly expressed in yeast. Yes, ChR2 is expressed but I disagree that it is “properly” expressed as most of ChR2 proteins not at the PM and so might not be functional.

(4) In the middle of page 10, the authors stated that the data is “not explicitly shown here”. I suggest reporting the results here or to remove this statement.

(5) The manuscript feels incomplete. The authors should consider performing a screen with their engineered yeast to demonstrate how it can be used and that it can be used successfully. Alternatively, the authors might want to consider characterizing ChR2 in yeast to show that yeast is a powerful vehicle to characterize a protein that will be difficult to study in its endogenous environment. In my opinion, the manuscript is partial without addressing this point.

Minor points to address

(1) Space missing between “Fig.” and numbers as well as between numbers and units (such as 1mM should be 1 mM) through the manuscript.

(2) Please indicate how many biological replicates for experiments in figure legends.

6. PLOS authors have the option to publish the peer review history of their article (what does this mean?). If published, this will include your full peer review and any attached files.

Reviewer #1: No

Reviewer #2: No

---

## [Author Response · Author response to Decision Letter 0]

5 Mar 2023

Reviewer #1: Major comments:

Reviewer: The functionality of the Channelrhodopsin in yeast was tested using a growth assay under different Na+ and K+ concentrations. These indirect results should be complemented with a direct measurement of channel functionality upon light stimulation.

Our response: The intention of this work was to create an easy test system for a high throughput screening of functional Channelrhodopsins. Our data clearly show in several complementary experiments that the different growth phenotypes of the yeasts can only be explained by the presence of an active blue light stimulated, retinal depending and cation conducting Channelrhodopsin. Our assay is even able to distinguish between two different forms of Channelrhodopsins (Fig. 5/Fig. 7). We don’t see what additional information we would obtain from difficult electrophysiological recordings of photocurrents. In fact the goal of the paper is to show that Channelrhodopsin activity can be assayed without measuring the currents in individual cells.

Reviewer: The growth curves shown in Figures 5 and 7 were performed using light at 500 nm. The authors mention that this wavelength was used to measure the yeast growth and activate the Channelrhodopsin. For yeast growth curves, measurements must be carried out at 600 nm. 

Our response: With all respect we disagree with the reviewer. The measurement of growth curves at 600 nm is a convention but this wavelength is from a physical point of view not obligatory. Indeed, there are many examples in the literature (see below) in which growth curves also from yeast were measured at 500 nm. In our experiments this is justified by the fact that we use the same light for stimulating the Channelrhodopsin and for measuring the OD. For readers used to OD600 measurements we have added the information that one OD500 unit is equivalent to 2.3 x 107 cells/ml.

References reporting yeast growth as OD500: Yeung et al. (1999) MBoC; Zühlke et al. (2016) Scientific Reports; Nilsson and Sunnerhagen (2011) RNA; Bernardo et al. (2014) FEMS Yeast Research. This list is not complete but contains Journals like FEMS Yeast Research, which are specialized in work with yeast. 

Reviewer: Therefore, the authors should use an illumination system coupled to a plate reader to stimulate the yeast cells at 460 nm and read the growth at 600 nm. Please see the illumination systems in the following references PMIDs: 27805047, 31788779, 34431694, and 36457859.

Our response: We do not see the need for implementing specific light sources. The intention of our paper is to create the biological basis for an easy functional test system. Every potential user can later employ our yeast strains for their specific need. This will eventually require anyway a specific adaptation like growth conditions, light source etc.

Reviewer: In Figure 5, the constant darkness condition (control condition) is missing.

Our response: The clear intention of this experiment is to analyze the negative effect of Na+ on yeast growth in the presence and absence of active Channelrhodosin. We don’t see a rational for comparing the growth data with dark incubated yeast. If the reviewer is concerned about some unexpected effects in the dark we can point to the experiments in Fig. 7. In these complementation experiments it was reasonable to include data on dark grown yeast. 

Reviewer: Figure 6 should be moved to supplementary information.

Our response: We have moved Fig. 6 to the supplementary information

Reviewer: In Figure 7 (panels B, C, and D), the statistical analysis should be done between constant darkness (control) and blue-light. 

In Figure 7 (panel A), the constant darkness condition (control condition) is missing.

Our response: Again, in both cases we don’t see the information which we would get from these experiments and comparisons with a dark control. Each experiment should address a clear cut question, which is answered by the results. A cross correlation of different conditions does not help to find these answers. 

Minor comments:

Reviewer: In table 1, replace “stuffer” by locus.

Our response: The term “stuffer” has been used in the original publication by Jansen et al, 2005, describing construction and use of the pGREG plasmids. For the sake of consistence with the original description we would like to keep this term.

Reviewer #2: Major points to address

Reviewer: (1) The first two figures contained failed attempt to expressed ChR2 in yeast, so the authors used a different approach in Fig. 3. Fig. 1 and 2 should be removed to the manuscript as they are distracting and not informative nor helpful for the rest of the data.

Our response: We understand the suggestion of the reviewer to start the manuscript with the successful part of the experiments. At the same time, we have valid arguments to keep the information presented in Fig. 1 and 2. A general problems is that negative results are rarely published with the effect that the same kind of experiments are repeated over and over by different laboratories. From conversations with colleagues, we think that the present paper is a good frame for reporting also the fact that external feeding of retinal is not sufficient for guaranteeing channelrhodopsin function in yeast. 

Reviewer: (2) The authors stated that N/K-ChR2-eYFP localizes at the plasma membrane (PM) with a subpopulation in endomembrane. However, it seems that the majority of ChR2 is at the vacuole with a small fraction at the PM (Fig. 5A). 

Our response: With all respect we have the impression that the reviewer misinterprets the anatomy of a yeast cell. Different from pant cells, the yeast vacuole is not filling up most of the inner space of the cell; it can be clearly distinguished from the plasma membrane. To illustrate this have included here two images with wt yeast cells (left) and with yeast cells expressing the GFP taged tonoplast channel YFC1. In both cases it clearly visible that the tonoplast is well separated from the plasma membrane. 

The images in our manuscript on ChR2 distribution in the yeast cells show definitely a plasma membrane localization of the protein. The two cells on the bottom-right show strong plasma membrane staining with nearly no other localization (the larger of the two cells) and staining associated with nucleus (ER) in the smaller of the two cells. We are now spelling this out in the revised manuscript and underpin that the functional data are fully supporting this conclusion. A growth complementation in low K+ medium by the expression of ChR2 in blue light can only be achieved by channel activity of this protein in the plasma membrane. 

Figures: Left: Transmission light image of wt yeast cells. Right: Overlay of transmission light image and fluorescent signal from GFP taged YFC1. 

Reviewer: It will suggest that most of ChR2 is non-functional nor folded properly. The authors should further characterize the expression and localization of ChR2 in yeast. ChR2 should mostly localize at the PM. Therefore, I disagree with the authors statement “When the optimized ChR2-5x, termed N/K-ChR2, was expressed in SHY4 cells, we found that it was also in this organism efficiently sorted to the plasma membrane”.

Our response: This criticism can be extended to any cellular system including neurons in which ChR2 and its variants are heterologously expressed (see for example: Mattis et al. (2014) Nature methods 11:763-772, Sugiyama et al. (2009) Photochem Photobiol. Sci. 8:328-336, Tomita et al. (2014) American Soc Gene Cell Therapy 22:1434-1440). Also in these systems the protein can be seen not only in the plasma membrane but also in endo-membranes. In fact, an improved sorting and targeting of ChR2 in mammalian cells is a big issue in the field of optogenetics. In addition to this general issue on ChR2 sorting it is important to remember that heterologous expression under a strong promotor will always generate some ChR2 (GFP fluorescence) in endomembranes. This originates from properly folded protein in the secretory pathway on the way to the plasma membrane or recycled protein in endosomes; also, a contribution of misfolded proteins cannot be excluded. In this respect our data are not different from results obtained with other expression systems. We now mention this in the manuscript.

Reviewer: (3) From Fig. 1 title, the authors stated that ChR2 is properly expressed in yeast. Yes, ChR2 is expressed but I disagree that it is “properly” expressed as most of ChR2 proteins not at the PM and so might not be functional.

Our response: See response to same criticism above. 

Reviewer: (4) In the middle of page 10, the authors stated that the data is “not explicitly shown here”. I suggest reporting the results here or to remove this statement.

Our response: We believe, the statement provides useful information. We decided not to add the 500 and 600 mM data in Fig. 5B, since this would make Fig.5B hard to read. However, the extracted data (reduced growth rate) for 500 and 600 mM NaCl are already given in Fig. 5C. They explicitly show that increasing NaCl concentration to 500 or 600 mM gives the same result as 400 mM and does not further augment the growth inhibition. Since the relevant data are included in Fig. 5C we delete the notion “not explicitly shown here” to avoid the impression that we are not showing the data.

Reviewer: (5) The manuscript feels incomplete. The authors should consider performing a screen with their engineered yeast to demonstrate how it can be used and that it can be used successfully. Alternatively, the authors might want to consider characterizing ChR2 in yeast to show that yeast is a powerful vehicle to characterize a protein that will be difficult to study in its endogenous environment. In my opinion, the manuscript is partial without addressing this point.

Our response: The intention of the manuscript is to provide an easy test or screening system for channelrhodopsins and even beyond this for any retinal dependent protein in yeast. We are not advertising this system for a detailed characterization of the proteins of interest. Such an in-depth characterization of channelrhodopsins still requires high resolution electrical or fluorescent methods. The title of the paper therefore reads ….for functional testing… To avoid wrong expectations in our system we are now spelling out in the revised manuscript in more detail the intentions, benefits and limitations of the system. 

Minor points to address

Reviewer: (1) Space missing between “Fig.” and numbers as well as between numbers and units (such as 1mM should be 1 mM) through the manuscript.

Our response: Thanks for pointing this out; we have corrected this.

Reviewer: (2) Please indicate how many biological replicates for experiments in figure legends.

Our response: We are surprised by this comment because the relevant numbers are already given in the figure legends where relevant, i.e. when statistical information is provided (Figs. 2, 4, 5 and 7)

---

## [Decision Letter · Decision Letter 1]

21 Mar 2023

PONE-D-22-35053R1Tailoring baker’s yeast Saccharomyces cerevisiae for functional testing of ChannelrhodopsinPLOS ONE

Dear Dr. Thiel,

Thank you for submitting your manuscript to PLOS ONE. After careful consideration, we feel that it has merit but does not fully meet PLOS ONE’s publication criteria as it currently stands. Therefore, we invite you to submit a revised version of the manuscript that addresses the points raised during the review process.

As you can see below, one review as a couple points that should be addressed before publication. I believe these can be easily addressed in a revised version of the manuscript. 

We look forward to receiving your revised manuscript.

Kind regards,

Patrick Lajoie, PhD

Academic Editor

PLOS ONE

Journal Requirements:

Reviewers' comments:

Reviewer's Responses to Questions

**Comments to the Author**

1. If the authors have adequately addressed your comments raised in a previous round of review and you feel that this manuscript is now acceptable for publication, you may indicate that here to bypass the “Comments to the Author” section, enter your conflict of interest statement in the “Confidential to Editor” section, and submit your "Accept" recommendation.

Reviewer #1: All comments have been addressed

Reviewer #2: (No Response)

2. Is the manuscript technically sound, and do the data support the conclusions?

Reviewer #1: Yes

Reviewer #2: Yes

3. Has the statistical analysis been performed appropriately and rigorously? 

Reviewer #1: Yes

Reviewer #2: No

4. Have the authors made all data underlying the findings in their manuscript fully available?

Reviewer #1: Yes

Reviewer #2: Yes

5. Is the manuscript presented in an intelligible fashion and written in standard English?

Reviewer #1: Yes

Reviewer #2: Yes

6. Review Comments to the Author

Reviewer #1: (No Response)

Reviewer #2: In this manuscript, the authors report an engineered and validated Saccharomyces cerevisiae strain that expressed the light-activated Channelrhodopsin 2 (ChR2) that could be used for future screens.

Overall, the authors addressed most of my concerned except the few points below that still need to be addressed.

(1) I understand the author response to my original comment (2). However, I am still unconvinced with the author explanation mostly because I am unable to confidently assess the localizations of N/K-ChR2-eYFP with the microscopy images shown in Fig. 5A. For most the cells, the protein seems to be localized internally but it can’t be determined in which organelle it is localized. Ideally, they should show the co-localization of N/K-ChR2-eYFP with organelle markers. Again, I have the impression that N/K-ChR2-eYFP is mostly internalized with a small proportion localized at the PM.

(2) Please indicate how many biological replicates for experiments in figure legends. The authors responded that “We are surprised by this comment because the relevant numbers are already given in the figure legends where relevant, i.e. when statistical information is provided (Figs. 2, 4, 5 and 7)”. I am sorry for the confusions, but I mean that it is not indicated in the method nor in figure legends if the other experiments were repeated at least in triplicates such as Fig. 1A, 1B, 1C, 1D, 2B (three values means three independent biological replicates?), 3A, 3B, 3C, 4C, 4D, 5A, and 6. Alternatively, the authors can state in the method section that all experiments have been conduced at least for three independent biological replicates if it is the case.

7. PLOS authors have the option to publish the peer review history of their article (what does this mean?). If published, this will include your full peer review and any attached files.

Reviewer #1: No

Reviewer #2: No

---

## [Author Response · Author response to Decision Letter 1]

27 Mar 2023

Dear reviewer, thanks for pointing out issues that might have been not clear yet in the previous version of the manuscript. We have now amended the manuscript accordingly:

Reviewer: 

I understand the author response to my original comment (2). However, I am still unconvinced with the author explanation mostly because I am unable to confidently assess the localizations of N/K-ChR2-eYFP with the microscopy images shown in Fig. 5A. For most the cells, the protein seems to be localized internally but it can’t be determined in which organelle it is localized. Ideally, they should show the co-localization of N/K-ChR2-eYFP with organelle markers. Again, I have the impression that N/K-ChR2-eYFP is mostly internalized with a small proportion localized at the PM.

Our response:

We do not question fluorescence in internal membranes, but GFP tagged channel protein is for sure visible in the plasma membrane of all cells, which express the channel protein. This occurs on a variable background ranging from no fluorescence signal inside the cell (cell now indicated in Fig. 5) to an intense labeling of internal structures. By adding two additional paragraphs we want to make the important point that already the presence of GFP fluorescence in the plasma membrane is sufficient for using the yeast system for complementation assays. This is irrespective of the presence/absence of cytosolic signals. 

To strengthen our message, we have:

a. added a quantitative analysis on fluorescence localization in the plasma membrane versus the cell interior. This analysis confirms the general presence of GFP fluorescence in the plasma membrane not only in a few images but in a large cohort of analyzed cells (page 5). 

b. We have added an experiment (page 3, Fig. S1) in which we confirm, with the help of a positive control, that a weak GFP signal from a channel protein in the plasma membrane is sufficient for using the K+ uptake deficient yeast strain for complementation assays. Such a complementation can only be interpreted in the context of channel activity in the plasma membrane. 

c. We have not performed colocalization assays since for our application it is not relevant to identify the cytosolic localization of the protein. From the biogenesis of a membrane proteins, we know that we will find positive colocalization with the Golgi and the ER. Although this information is interesting in the context of channel synthesis and sorting it will not provide crucial insights into the positive or negative outcome of our complementation assays. 

Reviewer:

Please indicate how many biological replicates for experiments in figure legends. The authors responded that “We are surprised by this comment because the relevant numbers are already given in the figure legends where relevant, i.e. when statistical information is provided (Figs. 2, 4, 5 and 7)”. I am sorry for the confusions, but I mean that it is not indicated in the method nor in figure legends if the other experiments were repeated at least in triplicates such as Fig. 1A, 1B, 1C, 1D, 2B (three values means three independent biological replicates?), 3A, 3B, 3C, 4C, 4D, 5A, and 6. Alternatively, the authors can state in the method section that all experiments have been conduced at least for three independent biological replicates if it is the case.

Our response: 

We included an extra chapter on “statistics” in the material and method section in which we state all the information on experimental replicates etc.

---

## [Editor Report · Decision Letter 2]

28 Mar 2023

Tailoring baker’s yeast Saccharomyces cerevisiae for functional testing of Channelrhodopsin

PONE-D-22-35053R2

Dear Dr. Thiel,

We’re pleased to inform you that your manuscript has been judged scientifically suitable for publication and will be formally accepted for publication once it meets all outstanding technical requirements.

Kind regards,

Patrick Lajoie, PhD

Academic Editor

PLOS ONE

---

## [Editor Report · Acceptance letter]

4 Apr 2023

PONE-D-22-35053R2 

Tailoring baker’s yeast *Saccharomyces cerevisiae* for functional testing of Channelrhodopsin 

Dear Dr. Thiel:

I'm pleased to inform you that your manuscript has been deemed suitable for publication in PLOS ONE. Congratulations! Your manuscript is now with our production department. 

Kind regards, 

on behalf of

Dr. Patrick Lajoie 

Academic Editor

PLOS ONE